# Secondary Education and Class Stratification: Understanding the Hierarchy of Sexuality Education in a Chinese Vocational High School

**DOI:** 10.3390/children9101524

**Published:** 2022-10-05

**Authors:** Chong Liu

**Affiliations:** School of Sociology and Social Policy, University of Leeds, Leeds LS2 9JT, UK; sscl@leeds.ac.uk

**Keywords:** social stratification, sexuality education, young people, education system, China

## Abstract

The discussion of sexuality education has a long history in China since the onset of modernisation in the early 20th century. Sexuality education has also existed in China’s educational system for a long time but in various forms. However, the discussions regarding students’ experiences and the influence of social stratification in China’s particular social context are still limited. From March to September 2019, the author visited an academic high school (*pugao*) and a vocational high school (*zhigao*) in Tianjin, China, to gain first-hand data to understand young people’s sexuality education experiences. In this paper, the author specifically paid attention to China’s social class, a rarely discussed topic in China’s society. She also tried to listen to the voices of young people and schoolteachers and help them to be heard. By presenting the insights from schoolteachers, students, as well as the materials from a nationally approved textbook, the author conducted a thematic analysis about how social class influences young people’s sexuality education experiences in secondary education in China.

## 1. Introduction

Up to now, in the international arena, when academic researchers and the public talked about sexuality education, the mainstream discussions have focused mostly on the successful experiences of the European and American countries, such as the UK, the USA, the Netherlands, Germany, and Denmark. This has been evident across a series of handbooks and edited volumes about experiences of sexuality education throughout the world [1,2,3,4]. However, there have been very few international discussions about sexuality education in developing countries and the Global South. Only in recent years an increasing number of studies have focused on those previously underrepresented countries. Considering the need for evidence-based sexuality education, it has become particularly important to fill the gap in international discussions and seek more global perspectives.

Comprehensive sexuality education, as a kind of evidence-based sexuality education, is now widely acknowledged as being the most effective and indispensable guide to help children and youth obtain systematic knowledge about sexuality [5,6]. Studies have indicated an understanding that comprehensive sexuality education should include not only discussions on biological reproduction but also other issues such as interpersonal relationships, sex identification, self-protection, gender equality, and diversity [7].

Whilst comprehensive sexuality education was first introduced in China in 2001, China has been largely underrepresented in academic discussion. Regarding the implementation of sexuality education in China, some scholars indicated that there was little sexuality education from 1949 to the late 1970s due to the strict political environment at that time [8]; yet, even in these cases, they rarely explained the exact definition of sexuality education in their contexts. Crucially, there are many variations in “sexuality education”, and comprehensive or evidence-based sexuality education is not the only available option. For example, sexuality education promoting abstinence has been playing an important role amongst children and youth in China’s society [5].

Education inequality is a long-lasting focus in China, typically the unequal distribution of educational resources is raising particular concerns amongst researchers and the public. Educational inequalities caused by geoeconomics and geopolitics throughout the country are the most notable issues regarding the unequal distribution of resources. Therefore, in many cases and related studies, when educational inequality is discussed, the debate is usually concentrated on the outcome of “economics” and “politics” in a broad sense. Regarding the capital theory raised by Bourdieu [9], it becomes reasonable to reflect on such inequality through consideration of forms of capital, specifically social capital, economic capital, and cultural capital. In the Chinese educational system—which is heavily immersed in traditional Chinese culture [10]—comprehensive sexuality education as a “superstructure” and challenge to this system can thus be discussed in tandem with the development of China’s society and long-lasting social inequality embedded in China’s specific context.

The main focus of this paper concerns institutional education stratification within high school education in China; namely, the paper explores differences between the academic pathway (*pugao*) and the vocational pathway (*zhigao*). In Woronov’s [11] book, *Class Work*, which explored China’s vocational education and young people’s experiences, she described China’s education system as a “class sorter”. Indeed, this class sorting dimension of education has hugely shaped young people’s expectations and their living experiences. It is notable to stress that social class in this paper has two different meanings. It does not only refer to individual’s social class status based on their capitals [9]. Particularly, in this research, it is more related to and defined by the institutional education stratification.

I had rarely thought about how social class and school types differentiate young people’s aspirations before I conducted my research, given my account of research motivations. As the only daughter of a middle-class family, being a successful academic has been a long-term goal throughout my life [12]. My best friends at high school all went to the best universities in China and later pursued their masters and Ph.D.s in different countries. Without knowing how education stratification plays its role in young people’s lives, I assumed that going through the academic pathway and attending university was part of the usual way to live my life. I did not understand that as being privileged given my background.

After beginning my research journey, I was surprised to realise that I did not know any vocational school students before I began my fieldwork. I recalled the years I spent in my primary and secondary schools. At the time when I was younger, I felt intimidated to fail. In my schoolteachers’ words, it was wholly unacceptable and shameful to fail at exams and to be stranded in the vocational pathway. It seemed essential to remain on the academic pathway, to achieve, and to be successful [12].

However, I was amazed by the beautiful, honest, frank, and kind souls that I met when I did my fieldwork at the vocational school. I started to wonder how and why they had been belittled so much. I also wanted to understand why, as the left-behind and the ignored in China’s education system, they could still have sexuality education in their Mental Health class and how much it was related to institutionally biased expectations.

Hence, in this paper, I first introduce the context and the related discussion about education stratification. Then, I unpack how high school education has been class-stratified. I will mostly focus on vocational school students’ experiences with sexuality education and highlight the differences and similarities between the existing form of sexuality education in vocational education and comprehensive sexuality education.

## 2. Context

Andreas [13] summarised that, before 1978, China’s education system, similar to its social structure, had been shaped as a pyramid, which had limited places at the top and a range of difficulties surrounding social mobilities. Even in the 2010s, education was still playing a vital role in social stratification [14]. In China, most educational institutions, such as primary schools, secondary schools, and universities, are funded by the central and/or local governments. In 1986, the concept of compulsory schooling (*yiwu jiaoyu*) was first brought into the Compulsory Education Law. This regulated that nine-year compulsory schooling should be delivered to all school-aged children regardless of their gender, ethnicity or race. Although there are some slight regional variations and exceptional cases in this implementation, most Chinese students normally spend 6 years in primary school (in some areas 5 years) and 3 years in middle school (in some areas 4 years) to complete their compulsory education. Afterwards, ideally, if they do not cease their studies, they spend another 3 years in academic high schools or in vocational high schools before taking university entrance examinations or entering the labour market [11]. Typically, those who “failed” to go to university would directly enter the job market and become workers in domestic labour-intensive industries or the lower end of the service sector.

This kind of stratification could actually be seen everywhere from preschool to postgraduate school in China. During the period of the Cultural Revolution from the 1960s to the 1970s, students’ and their families’ political status determined their social position and the educational resources they could access [14]. At the beginning of the post-Mao era, along with the re-emergence of the university entrance exam, education, as one of the most effective channels, helped many people realise their wishes for upward social mobility. Hence, education has become one of the most competitive battlefields in China’s society.

Subsequently, it is hardly surprising that in China nowadays, in addition to the exam result, the permission to go to “good schools”, or the academic pathway, in this case, could also be related to the social status of the students’ families. It can be related to their social capital, economic capital, and cultural capital, since the students’ exam results are partly influenced by their families’ investments in their study [14,15]. Furthermore, in present-day China, it is common to see that some parents do not hesitate to pay large amounts of money as “donations” in order to ensure that their children are accepted onto the academic pathway [16].

Lu [17] indicated that in China, social class could be another kind of reproduction with regard to the discussion about educational stratification, as it was highly dependent upon people’s family background. Therefore, this deep-rooted hierarchical education system has stratified an increasingly more unequal society [18,19]. Similarly, as Woronov observed [20], high school entrance exams could be viewed as a class sorter. Departing from these considerations, it is clear to a certain extent that the selection of schools is not only related to students’ family background but also plays an important role in their personal future and further social stratification in China.

The influence of social class on sexuality education in China emerged when it first came into China’s society in the early 20th century along with the New Culture Movement advocating democracy and science [21]. In Wang’s review of the history of sexuality education in contemporary China, as a kind of import, the idea of sexuality education was first brought back to China by well-educated, wealthy male intellectuals. Considering the patriarchal and hierarchical social circumstance in China’s society in general [22], it is thus clear to see the rationale behind this paper. In specific, the inequality and hierarchy have been rooted in sexuality education’s development in China.

At the same time, the interrelationship between sexuality and social class has been discovered in different social contexts. For example, in the book *Formations of Class & Gender* [23], Skeggs unpacked the cultural level of displaying class and the popular representations of class, especially the working class, in consideration of sexuality. According to her words, “the label working class when applied to women has been used to signify all that is dirty, dangerous and without value [23]”. Similarly, in Elley’s [24] work researching how social class had influenced British young people’s sexuality, she mentioned a stereotype in British society saying that working-class young men’s hobbies were “fighting, fucking, football” [25]. Elley also reckoned that it was essential to have sexuality education programmes to help reduce social inequalities and challenge these social stereotypes. However, this perspective is rarely discussed in China’s context. Some existing research mostly focused on the newly emerging middle-class social group [26,27,28]. Pan and Huang [29] revisited people’s sexuality activities, such as consuming sex work amongst different social class groups, through their nationwide survey. The qualitative investigation is still inadequate.

Regarding inequalities and stereotypes in sexuality education within the hierarchical Chinese educational system, the relationship between social class and educational resources offers a powerful and convincing argument. The following section outlines the research methods, which attributed great importance to educational inequalities in China when selecting the samples and sites for research.

## 3. Materials and Methods

Drawing a particular attention to educational stratification, the aim of this paper is to provide an analysis of how sexuality education in vocational high schools has been shaped and regulated by institutional social class stratification and biases. This paper is based on my Ph.D. research, which is a qualitative study exploring Chinese young people’s experience of sexuality education and their understanding of sexuality practice, identity, and relationships. I selected a group of high school students, aged between 16 and 18, as the primary participants for data generation. From March to September 2019, I conducted my fieldwork at a vocational high school as well as an academic high school in Tianjin, China, considering high school system as a class sorter [11].

There are multiple reasons for choosing Tianjin. First of all, it is because of the Nearby Enrolment Policy in Tianjin, advocated and partially practised from 2006 and fully implemented since 2015. In this case, students’ families have to consider the quality of education in the assigned schools based on their household registrations [30]. Therefore, schooling has become a privileged game of capitals. According to Li [19], the hierarchical education system has reproduced and boosted social inequality in China’s society. Another consideration for choosing Tianjin is because of this city’s specific societal and historical background. In her book *Hygienic Modernity*, Ruth Rogaski [31] used Tianjin as an example to explore how this city was influenced by the changing dynamic of the world order from the late 19th century to the early 20th century. Accordingly, Tianjin became an ideal case to help understand the diversity as well as the interrelation between the individuals and society in China, considering its specific social, cultural, political, and historical contexts.

During this period, I used qualitative research methods and carried out 35 semi-structured interviews with 28 student participants (14 young women and 14 young men) and 7 schoolteacher participants (5 women and 2 men) in order to understand the real-life experiences of sexuality education amongst young people in contemporary China. Since the interviews were semi-structured, the number of questions was not fixed. Apart from the introductory questions, I mainly focused on five main aspects, namely: formal and informal sexuality education; a sense of social norms; interpersonal relationships; masculinity and femininity; as well as sex, romance, and love. Interviews with student participants were carried out during their lunch break and dinner break. Interviews with schoolteacher participants were conducted when they were free during school time. All of the interviews were recorded after obtaining participants’ consent. Most of the interviews lasted 60–90 min.

I conducted a content analysis based on the data from in-depth semi-structured qualitative interviews and the Mental Health textbook. The textbook was one of the most significant teaching and learning materials that students and schoolteachers mentioned regarding young people’s sexuality education experiences at the vocational high school. The selection of content used in this paper was based on the consideration of the paper’s topic and aim. After reading the whole textbook, I chose some of the most relevant activities and content in the textbook as materials for further data analysis.

It is worth mentioning that in my research, although I explored the similarities and differences among a diverse group of young people, I did not wish to set the criteria to measure and judge these young people’s experiences. However, exploring the various factors that shaped their unique experiences was a fascinating avenue of analysis. In this analysis, I paid particular attention to institutional factors.

As Yang [32], Elley [24], Carmody [33], and other social scientists stressed in their studies on youth sexual learning, identity, and practice, it is crucial to listen to youths’ voices to understand their circumstances of sexuality and to foster the development of sexuality education. With this in mind, I decided to apply interpretivism as my epistemological standpoint to write for youth and make their voices heard by paying attention to “the sense people make of their own lives and experiences” [34]. In other words, in my research, youth, as the research subject, are people who hold the reality.

Youth participation can also be seen as a form of citizenship [35]. In my research, considering the future research impact, youth participation is rather important for future policy-making and related practices. Willow, Neale, and colleagues [36] proclaimed that citizenship is “an entitlement to recognition, respect and participation” to children and youth. Meanwhile, youth have the right to be seen positively and seriously as “competent social actors” [37,38]. Resonating with The Convention on the Rights of the Child, issued by the United Nations Children’s Fund in 1989 [39], all children should have rights and freedom to express themselves.

I prepared separate information sheets and consent forms for my student participants and schoolteacher participants due to some significant differences between the two versions. For example, in the information sheets, differences in the participant selection criteria and interview length were considered.

It is worth noting that in my research, I only obtained consent from my interview participants. This is because my participants, especially student participants, were all aged over 16 years old when I interviewed them. In China, by the General Rules of the Civil Law [40], my student participants already had limited civil conduct capacity, meaning they could decide whether or not to partake in my research. Moreover, in the UK, regarding the Statement of Ethical Practice issued by the British Sociological Association [41] and the University of Leeds Research Ethics Policy [7], generally youth over 16 years old can also give consent without their parents’ agreement.

Even though the group of young people in my research were not treated as a particularly vulnerable group, during my fieldwork, there were still some significant considerations. As Åkerström and Brunnberg [35] suggested, one of the challenges of researching with young people is that they may not fully understand the importance of the research process. With this in mind, I repeatedly informed them of how I would protect their privacy and the research data [42]. To be specific, some of my student participants were afraid that I would inform and disclose their “secrets” to their schoolteachers. In this case, I shared some of my feelings as if I was also a student, for example, that I would be someone who annoyed schoolteachers. Before I collected my research data, I spent quite a long time obtaining my participants’ trust. As time passed, they realised that I was a trustworthy secret keeper. Eventually, I was able to build rapport with my participants throughout my fieldwork.

All the participants have been given pseudonyms. Furthermore, regarding the consideration of anonymity beyond the assignment of a pseudonym, I also decided not to use the actual name of the schools.

## 4. Analysis

### 4.1. Individual Experience with Sexuality Education

There is a module named Mental Health designed explicitly for vocational high school students nationwide. Through my students’ introduction, there are many cases and units related to sexuality education in the textbook, such as sexual activity, sexual desire, and teenage pregnancy. This module is a significant part of the sexuality education experience amongst my student and schoolteacher participants at the vocational high school. For example, when I asked them if they had learned anything about sexuality according to their formal schooling, many students and Mental Health teachers I interviewed at the vocational high school mentioned this module and some content of this textbook to me.

Kangni was one of my schoolteacher participants at the vocational high school. She had taught Mental Health for a couple of years and explained:

We deliver this module to all second-year students in this vocational school. One class a week, that is all. […] I always try to deliver all important things mentioned in the text book to my students, but so rarely can I find students who listen to me carefully.(Mental Health teacher, female, vocation high school)

Kangni’s observation was reflected through students’ accounts in the vocational high school. Amongst all of my student participants at the vocational school, none of the year-three students mentioned the Mental Health module during the interviews. Only some of the year-two students described their memories and experiences of it. Therefore, it is possible to make speculations based on students’ responses—the Mental Health module did not leave a deep impression on them.

Jinru is another Mental Health teacher at the vocational high school, who had a bachelor’s degree in psychology. Moreover, she had less teaching experience than Kangni. She had less than two years’ teaching experience before fieldwork took place. She also shared her experiences of teaching the Mental Health module and some general feelings with me:

Most of time, I do not feel so good, because I cannot feel any sense of achievement. You know students here basically do not listen to you during the classes. When I first came to this school, I felt so disappointed. For example, when I taught, students they were just on their phones. Then my senior colleagues told me to get used to it.(Mental Health teacher, female, vocation high school)

With such ambition and expectation, Jinru said she did not enjoy her work so much, as she did not have enough subject-specific support. This is exemplified in the following quote:

I sometimes do not know how to teach. I wanted to follow the teacher’s guidance book, but they did not offer anything useful. There were only some activities from the textbook. Basically, I had to teach without any clear instructions. […] I remember there was also something about reproduction hygiene, but it was too rough. There were no instructions. How could I teach with so little support?(Mental Health teacher, female, vocation high school)

Both Jinru and Kangni emphasised their sense of powerlessness when they taught this Mental Health module. They were particularly unmotivated by students’ reactions. Especially under some strict restrictions due to political considerations, they could not use much information outside of the selected and censored teacher’s guidance book [43]. Even if they had the freedom to choose their teaching materials, limited access to the Internet due to the Great Firewall in China made it rarely possible to obtain the reliable and varied sexuality education materials from outside China [44]. In short, Jinru and Kangni did not have another option.

Echoing Jinru’s interpretation, it is also necessary to see her students’ responses. Momo was one of my student participants and was in Jinru’s Mental Health class. Like Jinru, Momo also felt that the Mental Health class was not adequate for helping her obtain sexuality education (although she believed sexuality education was very important). The module’s purpose was to enable students to have the capacity to deal with potential problems in their future lives. Momo’s quote is as follows:

Researcher: How did Jinru teach your Mental Health class?Momo: She was really shy. When she taught, she was smiling timidly. I could imagine how embarrassed she was.Researcher: How about your classmates’ reflection?Momo: They were laughing. They did not do anything special. Anyway, we are not going to listen carefully in any class. That was all.(17 years old, intermediate class, female, vocational high school student)

Dantong was Momo’s classmate in the vocational school—she was a 16-year-old young woman from a working-class family background. Dantong, similar to all other five female student participants (seven female student participants at the vocational high school in total) who were interviewed at the vocational high school, explained that she did not have any strong impressions from their Mental Health module, even though Jinru delivered the class only a few weeks ago. As mentioned above, the Mental Health module did not leave a lasting impression amongst any of the student participants at the vocational high school. Even though Dantong and Momo shared their memories about the module, it was still mostly about the surrounding circumstance, such as teacher’s and classmates’ reactions, rather than the knowledge and content.

Jinru admitted how timid she was. In the meantime, Momo noticed how embarrassed Jinru was. However, no one had tried to tell each other that this module was inadequate. The lack of impression and participation amongst students, referring to Jinru’s and Kangni’s description above, exposes the complexities in the classroom. Jinru’s and these young women’s experiences represent another common situation of sexuality education through schooling. In the end, students showed their displeasure at sexuality education experiences in schools. The insufficient communication between schoolteachers and students can deepen the schoolteacher’s misunderstanding of young people’s sexuality identity, practice, and relationships.

Apart from this Mental Health module, all of my participants mentioned their biology class in the second year of their middle school, when I asked them about their sexuality education experiences. Through their accounts, that was the first ever and the most impressive experience learning about sexuality within schooling. However, what they learned was more than the human reproductive system itself. Similar to the Mental Health module, it was also about people’s, especially their biology teachers’ and classmates’, reactions. Hence, taking the biology class in middle school as an example, Weihang described his experience carefully to me as the following quote shows:

When I was in middle school, the biology teacher taught about humans’ reproductive system. All of my classmates were very shy, so they, including me, laughed a lot. I did not listen to my teacher carefully. It was not that engaging. The teacher was very severe.(18 years old, working class, male, vocational high school student)

Through the similarities and differences between the biology class in compulsory middle school education and the Mental Health class in the vocational high school, people’s reactions and values regarding sexuality would play a more significant role than the knowledge in influencing young people’s experiences with sexuality education. Therefore, it is not surprising to see how sexuality education exists everywhere in young people’s lives and how institutional factors should be discussed with regard to young people’s sexuality education experiences.

### 4.2. Social Class and Expectation

The discussion regarding social class and stratified value and expectation has attracted researchers’ attention worldwide from various perspectives over the last century. For example, Willis [45], in his book *Learning to Labour*, through an in-depth ethnographical investigation of a group of working-class young men in a secondary school in England, illustrated that youth’s values and expectations are strongly affected by their social class. Likewise, the discussion about value and expectation among youth has also been increasingly popular in Chinese academia. For example, Fong [46], in her book *Only Hope*, discusses China’s urban only-child generation and their expectations, indicating that this was set as a default option to encourage youth to pursue a “first-world” life. Similarly, Woronov [11], in her book *Class Work*, which explores vocational school students’ experiences in contemporary China, also demonstrated that no matter what kind of family background youth are from, the willingness to live a better life could not be more apparent in their everyday lives. These values and expectations come not only from parents but also from wider society. Even though class plays a vital role in people’s everyday lives in China’s society, class dimensions are usually ignored in related academic discussions [47].

In my research, according to the interview data, I evaluated participants families’ consumption level as economic capital, their parents’ educational qualifications as cultural capital, and their parents’ occupation as social capital. Accordingly, amongst 13 academic high school student participants, 7 (53.8%) were from middle-class backgrounds. Only two (15.4%) were from an intermediate-class background and four (30.8%) from a working-class background. However, at the vocational high school, amongst 15 student participants, none (0.0%) were from a middle-class background. There were six (40.0%) from an intermediate-class background and nine (60.0%) from a working-class background. Hence, to a certain extent, the statistics demonstrate how China’s high school education system is class-stratified [11].

Echoing the high aspiration amongst academic high school students, Songli’s interpretation can be seen as a realistic insight from the schoolteacher’s perspective. Her quote is as follows:

Our school hopes that studens do not take part in romantic relationships. It is because we are afraid that if they break up, it would then have a negative impact on their academic performance before the university entrance examination. Thinking too much about things other than study, such as sexuality-related things, would affect their examination results and ruin their future due to the mood swings […]. Besides, it might also endanger their safety. What if they get pregnant or make others pregnant? How could we explain to their parents?(Mental Health teacher and moral education officer,female, academic high school)

Since the economic reforms in 1978, sexuality education has been officially promoted and non-officially advocated through various policies and laws. For instance, from the late 1990s, some universities in China launched sexuality education courses [48]. However, and as represented by Songli’s quote, most adults, such as schoolteachers and policymakers who control educational provision, were mainly born and raised during the revolutionary era (from 1950s to 1970s). These people tend to worry about the popularisation of sexuality education. They believe that once children receive knowledge about sexuality, they will engage in sexual activities [29]. This kind of thought is influenced by conventional values towards sexuality, which were developed during conservative times due to the dual-modernity model proclaimed by Liu [49]. Specifically, along with China’s social transformation, China’s cultural and economic development do not advance at an equal pace. Regarding its cultural modernisation, China’s cultural nationalism still plays a significant role [22].

For further discussion about interrelationships of sexuality education, school types, and aspiration, I thus want to stress the hierarchy of China’s education system. After compulsory education, middle school graduates choose and are chosen by their high school entrance examination (HSEE) results to continue their study in the academic pathway or vocational pathway [20]. Due to the prejudice targeting vocational school students, they are usually classified under the “failed” category to differentiate them from their peers in the academic pathway. Woronov [20] argues that HSEE can be seen as a class sorter. Echoing China’s central government’s requirement for fulfilling the high demand of the labour force, by launching policy, nearly half of the middle school graduates fail their HSEE nationwide [11]. This means that half of the middle school graduates are removed from further academic competition. Specifically, these students do not take the university entrance examinations and have fewer chances to obtain professional white-collar jobs, thus having minimal opportunity to realize upward social mobility. This also demonstrates how inequality and social stratification are reproduced within China’s education system.

For Brine [50], social class is usually regarded as one of the most essential factors to focus on in education-related discussion. It is almost impossible to understand the complexity of sexuality education and social stratification without thinking about the educational system in China, as it is highly class-stratified itself [11]. Research shows that both individual and institutional/group meritocratic expectations matter [50,51]. Social stratification in schools, different from parenting, peer communication, and media, refers to both the macro-level class, namely school type, as well as the micro-level class, measured by various capitals, as shown in my following analysis. Gender also plays a vital role in education [52]. In my research, school type can be viewed as an efficient lens to understand youths’ values and expectations regarding class and gender and a starting point for discussion surrounding sexuality education and social stratification. To illustrate this point, Songli’s words mentioned above show how the “good students” are expected to have a “bright future” by using the exam-oriented education system as a façade to rationalize the insufficiency of sexuality education. Considering the above discussion, the division between the academic pathway and the vocational pathway is clear.

As highlighted earlier in this paper, Jinru and Kangni, my schoolteacher participants who were both Mental Health teachers at the vocational school, told me that they were teaching sexuality-related topics. These topics included gender inequality, intimacy, romantic relationships, unintended pregnancy, masturbation, sexual desire, and some other issues in the Mental Health module. Their teaching activities were based on the teaching and learning materials named *Mental Health* [53], which are designed specifically for vocational school. Higher Education Press, a professional publishing house directly affiliating with the Ministry of Education of the People’s Republic of China, publishes the textbook. The book includes some specific cases about student–teacher romantic relationships on page 43. There is also some discussion around unintended pregnancy and virginity on page 50. However, the value delivered by the curriculum is still relatively conservative. In the very beginning of the revised (3rd) version, it states:

This revision aims to implement the spirit of the 19th National Congress of the Communist Party of China, based on the Xin Jinping Though in the Chinese Characteristic Socialism for a New Era. It serves the vocational school students’ mental health education, and helps students understand and deal with the mental health and behavioural problems they will face during their growth, studying, living, job-seeking, and working [53].

Vocational school students, in this case, have been uniquely highlighted for the potential problems they may encounter in the future. On pages 50 and 51, there are some examples from the *Mental Health* textbook that are designed explicitly for vocational school students:

Activity 1: DiscussionWeiwei had her surgical abortion due to the unintended pregnancy. How would she be harmed physically and psychologically? Activity 2: Water ExchangeTeachers should have prepared several glasses of coloured water in advance. You can choose one colour and exchange water with others—give a small amount of water to others, and get a small amount of water from others in different colours. Please observe the changes in water in your glass.You can have a look at the colour of water in your glass. How do you feel?Have you gained any inspirations?How would you deal with your sexual ideas and sexual desire in the future? 

Activity 1 concerning abortion reflects a debate regarding its advantages in recent decades worldwide. For example, researchers [54,55,56] argue that abortion can be viewed as a kind of empowerment to women. For young women especially, they do not have to start motherhood early and suffer the negative consequences. Abortion could be an option to re-start their new lives, return to education, and enhance their subjective well-being. However, it is clear that the discourse in the teaching and learning materials designed specifically for vocational school students still conservatively highlights only the negative aspects. Teaching thus becomes an act of threatening.

Furthermore, Activity 2 tells students that sexual contact will make them “dirty” while using water as a metaphor for chastity, as the result of the activity is relatively straightforward because the water’s colour gets darker when water from the different glasses is mixed. On page 49, Activity 2 intends to foreground the importance of students’ chastity as is indicated earlier: “If you do value the affection, first of all, cherish your own body”, and “Cherish yourselves like the way you cherish pure water. Do not lose yourselves, and make your youth beautiful forever!” 

Echoing Skeggs’s [23] words saying working-class women’s sexuality was usually labelled “dirty, dangerous and without value”, the design of Activities 1 and 2 could be seen as a kind of reproduction of stereotypes and a reinforcement of the sexual hierarchy. Therefore, the design of the *Mental Health* textbook is an outcome of the institutional discrimination against vocational school students along with the stratification within China’s education system since its reformation about half a century ago. From a more institutional perspective, the school’s category becomes a field for shaping the youth sexuality education experiences.

When I asked about the youth’s aspirations during interviews, vocational school students described how they belittled themselves according to what society *constructed* them to be. As explained by nearly all my schoolteacher participants, in both academic and vocational schools, vocational school students start their socialisation much earlier than their peers in the academic pathway. As such, they assumed that vocational school students would have many more opportunities to have sex, or “troubles”, than others. To some extent it is positive that the government has begun to encourage high schools to educate their students about sexuality. However, it is still highly conventional and oppressive. The widespread stereotype of seeing vocational school students as potential “trouble makers” can leave schools and students divided into different levels and deepen social stratification in education. With such discrimination and inequality, it is hardly surprising to see the arduousness of implementing sexuality education systematically in the education system.

Schoolteachers and students themselves highlight the importance of academic competence, even if at the vocational school which is constructed as a field where academic performance is less crucial. Echoing the education policy in China, around half of the middle school graduates have to go to vocational schools. To some extent, vocational school students are *made* to be vocational school students by society. As a result, institutionally, youth sexuality education experiences can be severely influenced by their class-determined school types and differentiated aspirations.

## 5. Conclusions

At the beginning of this paper, I unpacked sexuality education in China’s context. Through the analysis above, it is not difficult to see that sexuality education exists in China. Most importantly, sexuality is not just something young people learn about in school. Rather, it is more like air—it exists everywhere and is a part of everyone’s life. The cultures, politics, practices, and norms that exist inside and outside young people’s classrooms have heavily influenced young people’s sexuality education experiences.

In my Ph.D. research, I raise the concept of “sexuality education with Chinese characteristics” to explain the institutional barriers to implementing evidence-based sexuality education and draw upon the youth sexuality education experience from personal and institutional perspectives. This is one of the possible ways to understand why vocational school students have to experience the class-determined projection in this “class sorter” [11]. The practice of this class-stratified education system also confirms the rejuvenation of Confucianism after the implementation of economic reformation in 1978 to rationalize social inequalities in China’s society [22]. In this case, Confucianism became an efficient tool to advocate a hierarchical social system. Therefore, it is reasonable to assume that traditional values still play a significant role in China’s society today.

However, the above consideration of comprehensive sexuality education and evidence-based education are far lagging in China [7]. There is an urgent need to advocate and develop a sexuality education system that can help reduce social inequalities and challenge these social stereotypes [24]. China’s uniqueness was highlighted many times by both schoolteacher participants and student participants when they talked about implementing sexuality education. Similar to the famous argument about the “Chinese characteristic”—whereby the government-advocated “socialism with Chinese characteristics” was like a protective cover for keeping space for further interpretation and mediation [57]—to some extent, “China’s uniqueness” needs more discussion and exploration.

This study is limited, in that this research was done in only two high schools in Tianjin. For future research, some more discussion about the diversity of school types could be considered. To be specific, there can be some other research projects focusing on schools in remote areas and elite international schools in China. Therefore, young people’s experiences with sexuality education could be explored in a more multi-dimensional way. Moreover, I would like to appeal to academics to pay more attention to the marginal groups such as vocational school students and teachers and their unheard voices, so that they can be seen and heard.

## Data Availability

The data presented in this study are not publicly available due to ethical considerations (to protect the privacy of participants).

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
