# Peer review of "Secondary Education and Class Stratification: Understanding the Hierarchy of Sexuality Education in a Chinese Vocational High School"

_children, 2022, doi:10.3390/children9101524_

Round 1
Reviewer 1 Report
Dear Author,
Thank you for bringing this sensitive topic to readers. The introduction and context that you have given in your paper, is quite comprehensive and clear, but the methodological part, analysis and conclusion part need to be developed and made stronger. Please see some comments in the document.

Author Response
Thanks a lot for your comments and suggestions. Please check the attached document for my responses.

Reviewer 2 Report
The article addresses an area that is definitely underrepresented in the current literature. The literature review focuses on sex education in China, which makes sense, but because the role of social class emerges as an important factor, it would be helpful to include additional references related to social class and sex education. There are a number of places where subjects raise issue about the lack of student involvement in the vocational schools. It would be nice to have information on what (if anything) the students learned from the lessons in school. Given the final argument in the paper, it would also be helpful if the literature review contained some background on stereotypes of working class sexuality. It seems like the students at the vocational school are being pushed towards a fulfilling a stereotyped view of lower class sexuality. Since there is so much overlap with stereotypes of working class sexuality (in Europe/North America), additional discussion of the (imagined) relationship between class and sexuality is definitely warranted.
Author Response

(The authors gave the same response as above.)
